# p97/UBXD1 Generate Ubiquitylated Proteins That Are Sequestered into Nuclear Envelope Herniations in Torsin-Deficient Cells

**DOI:** 10.3390/ijms23094627

**Published:** 2022-04-21

**Authors:** Sarah M. Prophet, Brigitte S. Naughton, Christian Schlieker

**Affiliations:** 1Department of Molecular Biophysics & Biochemistry, Yale University, New Haven, CT 06520, USA; sarah.prophet@yale.edu (S.M.P.); brigitte.naughton@yale.edu (B.S.N.); 2Department of Cell Biology, Yale School of Medicine, New Haven, CT 06520, USA

**Keywords:** dystonia, DYT1, TorsinA, p97, UBXD1, YOD1, ubiquitin, ERAD, Ufd1/Npl4

## Abstract

DYT1 dystonia is a debilitating neurological movement disorder that arises upon Torsin ATPase deficiency. Nuclear envelope (NE) blebs that contain FG-nucleoporins (FG-Nups) and K48-linked ubiquitin are the hallmark phenotype of Torsin manipulation across disease models of DYT1 dystonia. While the aberrant deposition of FG-Nups is caused by defective nuclear pore complex assembly, the source of K48-ubiquitylated proteins inside NE blebs is not known. Here, we demonstrate that the characteristic K48-ubiquitin accumulation inside blebs requires p97 activity. This activity is highly dependent on the p97 adaptor UBXD1. We show that p97 does not significantly depend on the Ufd1/Npl4 heterodimer to generate the K48-ubiquitylated proteins inside blebs, nor does inhibiting translation affect the ubiquitin sequestration in blebs. However, stimulating global ubiquitylation by heat shock greatly increases the amount of K48-ubiquitin sequestered inside blebs. These results suggest that blebs have an extraordinarily high capacity for sequestering ubiquitylated protein generated in a p97-dependent manner. The p97/UBXD1 axis is thus a major factor contributing to cellular DYT1 dystonia pathology and its modulation represents an unexplored potential for therapeutic development.

## 1. Introduction

Torsins are essential [1] AAA+ ATPases that localize within the endoplasmic reticulum (ER)/ nuclear envelope (NE) membrane system [2,3], where they carry out poorly understood functions. Torsins are unusual AAA+ ATPases as they strictly require interactions with one of two transmembrane activators to hydrolyze ATP [4,5,6]. The first of these activating proteins is lamina-associated polypeptide 1 (LAP1) and the second is luminal domain-like LAP1 (LULL1) [7]. Mutations within the C-terminal domain of TorsinA [8], one of the four Torsins encoded by the human genome [9], cause the neurological movement disorder DYT1 dystonia [2,10,11]. These DYT1 dystonia-causing mutations disrupt interactions between TorsinA and LAP1 or LULL1 [4,8,12], demonstrating the importance of the Torsin/activator complex during neurological development.

While the precise biological role of Torsins remains unknown, herniations of the NE are observed across all Torsin loss-of-function models ranging from *Caenorhabditis elegans* [13] to mouse neurons [1] (see schematic illustration, Figure 1A). These herniations, referred to as blebs, represent aberrant nuclear pore complex (NPC) biogenesis intermediates that are arrested prior to inner and outer nuclear membrane fusion [14,15,16]. Although blebs do not harbor mature NPCs, specific nucleoporins (Nups) associated with building multiple subcomplexes of the NPC are found inside these herniations [14,15,17,18]. In addition to Nups, the protein of unknown function myeloid leukemia factor 2 (MLF2) localizes to blebs [15,18] by a yet-to-be identified mechanism.

Beyond Nups and MLF2 [15,18], blebs are also highly enriched for components that suggest a protein quality control (PQC) defect exists in Torsin-deficient cells. First, blebs harbor Lys48-linked polyubiquitin (K48-Ub) chains [17,19] conjugated to proteins of unknown identity or origin. As K48-Ub is canonically associated with proteasome-mediated degradation [20], its enrichment inside blebs indicates that Torsin-deficient cells may have a compromised PQC mechanism. Secondly, blebs sequester a specific chaperone network composed of abundant HSP70 and HSP40 members [18]. The presence of HSP70s and HSP40s inside blebs along with K48-Ub suggests that a stress-related chaperone activity such as facilitating protein degradation [21] may also become dysregulated in Torsin loss-of-function models.

As neither the identity nor the origin of the K48-Ub protein inside blebs is known, the PQC pathway that becomes dysregulated upon Torsin deficiency remains poorly understood. Consequently, whether and how this PQC pathway contributes to DYT1 dystonia remains unknown. K48-Ub accumulation in blebs could indicate that proteins become ubiquitylated in response to Torsin dysfunction or that proteins fail to be degraded when Torsin function is compromised. While DYT1 dystonia is the most common congenital form of dystonia [10], no curative treatment has been reported and therapeutic options are only partially effective. Thus, a more accurate characterization of the PQC defect in Torsin-deficient cells may lead to the identification of yet unexplored pathways and define molecular players that might represent new targets for therapeutic intervention.

In this report, we employ a Torsin-deficient cell line as a model system to scrutinize the poorly understood process of K48-Ub accumulation widely observed in disease models of DYT1 dystonia. We demonstrate that p97 activity is required to generate K48-Ub accumulation inside blebs. We further show that K48-Ub conjugates inside blebs are unlikely to result from de novo synthesis or canonical ER-associated protein degradation (ERAD) as depleting components of the Hrd1–Ufd1/Npl4/p97 axis has little effect on K48-Ub deposition in blebs. A candidate approach aimed at the identification of alternative p97 cofactors revealed that the incompletely understood p97 adaptor UBXD1 and the deubiquitylating enzyme YOD1 are required for Ub accumulation. Stimulating an increase in global ubiquitylation by heat shock results in a stark increase in K48-Ub conjugates sequestered into blebs. This supports the idea that NE blebs have a remarkable capacity for sequestering misfolded, ubiquitylated species and preventing their degradation.

Taken together, our observations reveal that the p97/UBXD1 axis is a central player for the cellular pathology caused by defects in Torsin function. As p97 is a druggable target for a diverse set of diseases [22], its potential involvement in DYT1 dystonia reveals unexplored therapeutic opportunities to improve the lives of patients suffering from this debilitating disease.

## 2. Results

### 2.1. p97 Activity Is Required for K48-Ub Accumulation inside NE Blebs

The cellular processes that produce the K48-Ub conjugates found inside blebs (Figure 1A) have not been identified. One essential K48-Ub-directed enzyme in mammalian cells is the p97 ATPase (also called VCP in mammals and Cdc48 in yeast). While p97 activity functions in a number of diverse processes, its best characterized role is to mobilize ubiquitylated clients from membranes, interaction partners, or aggregates [23]. After extraction, these ubiquitylated proteins are often degraded by the proteasome. Because p97 is a major K48-Ub-directed enzyme, we investigated whether its activity was relevant for generating the K48-Ub conjugates inside blebs.

To inhibit p97, we treated TorsinKO cells with the p97 inhibitor CB-5083 [24,25]. As MLF2-HA localizes to blebs in a ubiquitin-independent manner [15], this construct was expressed to distinguish TorsinKO cells with blebs from those without. After four hours of p97 inhibition, the K48-Ub signal inside blebs was significantly depleted (Figure 1B,C) despite MLF2-HA remaining efficiently sequestered inside blebs (Figure 1B). To determine whether p97 is functioning directly within the bleb, we assessed its localization. Consistent with our previous observations [15,17], we found it was not stably enriched within blebs (Figure 1D) although we cannot exclude the formal possibility that p97 is present at levels below the detection limit of immunofluorescence (IF). While it is possible that p97 acts directly within the bleb lumen to generate the K48-Ub proteins, p97 activity at locations distinct from the bleb may also be required for K48-Ub accumulation in Torsin-deficient cells.

### 2.2. p97 Associates with More K48-Ub in Torsin-Deficient Cells

Upon finding that p97 activity is important for accumulating K48-Ub conjugates inside blebs, we asked whether p97 is generally associated with more ubiquitin in TorsinKO cells compared to the wild type (WT). To address this, we prepared soluble detergent extracts and performed an immunoprecipitation (IP) under native conditions with an anti-p97 antibody. Upon immunoblotting the IP with an anti-K48-Ub antibody, we observe that endogenous p97 is associated with significantly more K48-Ub in TorsinKO cells compared to WT (Figure 1E). This suggests that cells with deficient Torsin ATPase activity have more ubiquitylated p97 clients than cells with functional Torsins.

Taken together, we demonstrate that p97 associates with K48-Ub to a greater extent in TorsinKO cells compared to WT and that p97 activity is required for these ubiquitylated proteins to localize within blebs.

### 2.3. p97 Does Not Require the Ufd1/Npl4 Heterodimer for the Majority of the K48-Ub Protein Deposition to Blebs

The increased association of p97 with K48-Ub in TorsinKO cells suggests that a process linked to p97 function becomes perturbed during Torsin deficiency. One major p97-dependent process is ERAD [26]. During ERAD, p97 interacts with the Ufd1/Npl4 heterodimer to extract ubiquitylated glycoproteins from the ER membrane [27]. To determine whether ERAD-directed adaptor proteins contribute to K48-Ub accumulation within blebs, we depleted the heterodimer by treating cells with Ufd1/Npl4 targeting siRNA for 48 h (Figure 2A, cf Figure 3C). Knocking down Ufd1/Npl4 caused a minor decrease in the number of K48-Ub foci around the nuclear rim of TorsinKO cells (Figure 2A,B). However, this effect was far less pronounced compared to the K48-Ub depletion observed upon inhibiting p97 activity (Figure 1B,C). This suggests that a Ufd1/Npl4-independent p97 activity generates the majority of the K48-Ub protein inside blebs.

To validate that knocking down Ufd1/Npl4 resulted in measurable functional defects, i.e., stabilizing substrates destined for degradation, we expressed the short-lived protein LBR-1600* [28] (Figure 2C). LBR-1600* is a mutant of the ER- and inner nuclear membrane-localized lamin B receptor that is degraded in a p97- and proteasome-dependent manner [28,29]. Upon knocking down Ufd1/Npl4, LBR-1600* was strongly stabilized to the same extent as co-overexpressing a dominant-negative p97 construct, p97-QQ (Figure 2C). We therefore conclude that depleting the Ufd1/Npl4 heterodimer substantially stabilizes ERAD substrates but does not deplete blebs of K48-Ub conjugates to a similar extent as inhibiting p97.

### 2.4. Many Canonical ERAD-Associated E3 Ligases Do Not Significantly Contribute to the K48-Ub Protein inside Blebs

Prior to cytosolic degradation, ERAD substrates must be ubiquitylated and (in some cases) translocated across the lipid bilayer of the ER [26]. Ubiquitylation occurs on the cytosolic face of the ER and can be achieved by several E3 ligases embedded within the ER membrane. These E3 ligases not only ubiquitylate ERAD substrates but can also serve as dislocon channels within the ER membrane [30,31,32]. ERAD substrates are primed with ubiquitin by these E3 ligases so that the p97 machinery can recognize and extract these clients [27,33]. Depending on the nature of the ERAD substrate [34,35,36], the dislocon that enables passage out of the ER is canonically formed by either of the two highly conserved complexes composed of Hrd1 [30,37] or Doa10 [32] (MARCH6 in mammalian cells). While lower eukaryotic ERAD relies mostly on these two complexes, metazoans have alternative E3 ligases and mechanisms such as gp78 [26,36].

To investigate the involvement of the canonical E3 ligases in K48-Ub bleb accumulation, we depleted Hrd1, MARCH6, or gp78 from TorsinKO cells for 48 h (Appendix A). None of these conditions significantly affected K48-Ub accumulation inside blebs to the extent achieved by p97 inhibition (compare Appendix A and Figure 1B). Taken together with the minimal to modest effect of Ufd1/Npl4 depletion described above (Figure 2A, B), these data suggest that p97 does not require canonical ERAD machinery to generate the K48-Ub proteins sequestered inside blebs. These results further suggest that if ERAD substrates contribute to the K48-Ub inside blebs, they are likely retrotranslocated and ubiquitylated by a noncanonical mechanism largely independent of Hrd1, MARCH6, and gp78.

### 2.5. Newly Synthesized, Misfolded Proteins Do Not Account for the Majority of the K48-Ub Protein inside Blebs

An estimated 30% of all newly synthesized proteins are defective and degraded by the proteosome [38]. Newly synthesized misfolded proteins thus represent a constant threat to cellular proteostasis [39]. For this reason, cells have evolved multiple mechanisms to prevent translational errors and degrade aberrant proteins [40]. As the process of translation produces a significant number of ubiquitylated polypeptides, we determined whether newly synthesized ubiquitylated proteins contribute to the K48-Ub signal inside blebs. To this end, we monitored K48-Ub accumulation in blebs when translation was inhibited via cycloheximide (CHX) treatment. To confirm the efficacy of inhibiting translation, we performed a CHX chase in WT HeLa cells expressing the short-lived protein HA-LBR 1600* [28]. As expected, CHX treatment resulted in a steady decline of HA-LBR 1600* over time, consistent with a blockage of de novo synthesis and concomitant proteasomal turnover (Figure 2D). When translation was inhibited in TorsinKO cells by CHX treatment for up to seven hours, we observed no change in the accumulation of K48-Ub protein within the bleb (Figure 2E). As a four-hour treatment with the p97 inhibitor CB-5083 results in a near-complete depletion of K48-Ub inside blebs (Figure 1B,C), this suggests that newly synthesized misfolded proteins are unlikely to significantly contribute to the K48-Ub cargo sequestered into blebs.

### 2.6. The Relevant p97 Activity for Accumulating K48-Ub inside Blebs Depends on the Cofactors YOD1 and UBXD1

p97 is functionalized by at least 40 different interaction partners in mammalian cells [41] that direct its ATPase activity to distinct processes including ERAD, membrane fusion, and lysosome clearance [41,42]. To better understand what cellular process(es) requires p97 activity to produce the K48-Ub conjugates inside blebs, we determined the effect of depleting specific p97-interacting proteins.

While p97 interacts with many cofactors/adaptors, at least three major p97 complexes exist. These are the Ufd1/Npl4 heterodimer, p47, and UBXD1 [41]. The Ufd1/Npl4 dimer is canonically associated with recruiting p97 to the ER membrane during ERAD [43] whereas p47 functionalizes p97 during homotypic membrane fusion [44]. UBXD1, however, has been reported to contribute to a number of processes including trafficking events [45,46], autophagy of lysosomes [47], mitophagy [48], and ERAD [49]. Beyond these three complexes, p97 requires the deubiquitinase (DUB) YOD1 to participate in a number of processes including ERAD [50] and lysosome clearance [47].

To interrogate the diverse processes to which p97 contributes, we depleted p47 and UBXD1 by RNAi in TorsinKO cells (Figure 3A–C). To monitor bleb formation, these cells expressed MLF2-HA (Figure 3A). To perturb YOD1 activity, we expressed WT or a dominant-negative YOD1 construct YOD1-CS-FLAG [50] in TorsinKO cells co-expressing MLF2-HA (Figure 3A,B).

Upon depleting UBXD1 or perturbing YOD1 activity, the number of K48-Ub foci around the nuclear rim was significantly decreased (Figure 3A,B). This was not the case when p47 was depleted (Figure 3A,B). As WT YOD1 is an active DUB, expressing the WT construct also resulted in a significant decrease in the number of K48-Ub foci in TorsinKO cells (Figure 3A,B). This suggests that ubiquitylated proteins targeted by YOD1 localize to blebs as removing the ubiquitin modification by overexpressed WT YOD1 also depletes blebs of K48-Ub (Figure 3A,B). Taken together, these results suggest that the p97 interactors UBXD1 and YOD1, but not p47, participate in the process(es) that generate the K48-Ub proteins inside blebs.

MLF2, a protein of unknown function, is highly enriched inside the lumen of blebs [15,18]. While UBXD1, WT YOD1, and YOD1-CS are not enriched inside blebs (Appendix A), MLF2 may contribute to bleb formation by interacting with these p97 adaptors elsewhere within the cell. To test this possibility, we conducted an IP with an anti-FLAG antibody from TorsinKO cells expressing FLAG-tagged WT YOD1, YOD1-CS, UBXD1, or AIFM1 (Figure 3D). AIFM1 is a known MLF2-interacting protein [51,52] and therefore served as a positive co-IP control (Figure 3D). We found that MLF2-HA specifically co-immunoprecipitated with UBXD1 and WT but not dominant-negative YOD1 in TorsinKO cells (Figure 3D). While these interactions may be direct or indirect, MLF2 may function with specific p97 complexes during the generation of ubiquitylated protein and rely on active YOD1 for recruitment to these sites.

### 2.7. Provoking an Increase in Global Ubiquitylation Causes More K48-Ub Conjugates to Become Sequestered inside Blebs

Whether blebs are enriched for specific K48-ubiquitylated proteins or sequester general ubiquitin cargo is not understood. While we have performed multiple mass spectrometry-based workflows in an attempt to identify the K48-Ub conjugates inside blebs [15,18], we have not observed the enrichment of a specific set of ubiquitylated proteins. This suggests that blebs may possess an extraordinarily high capacity for sequestering a wide variety of ubiquitylated species, and the enrichment of specific proteins may therefore be difficult to capture by mass spectrometry. Thus, to probe the capacity of blebs to sequester ubiquitylated protein, we subjected WT and TorsinKO cells to heat shock, a condition that is known to greatly increase the amount of ubiquitylated protein inside cells [53,54,55].

Upon subjecting TorsinKO cells to 42 °C for 16 h, the amount of K48-Ub inside blebs increased significantly (Figure 4A–D). This effect was observed by IF (Figure 4A,B) and by biochemical fractionation wherein cells were enriched for ER/NE fractions (Figure 4C,D). This enrichment involved isolating nuclei and membrane fractions by centrifugation through a sucrose gradient. ER/NE fractions were further enriched by centrifugation following DNase/heparin treatment. Successful fractionation was confirmed by immunoblot using antibodies against emerin (enriched in the ER/NE) and hnRNPA1 (enriched in the nucleoplasmic fraction) (Figure 4C). After validating successful fractionation, the ER/NE samples were subjected to immunoblot with an anti-K48-Ub antibody (Figure 4D). While WT HeLa cells had approximately the same amount of ER/NE-associated K48-Ub during unstressed (37 °C) and stressed (42 °C) conditions (Figure 4D), TorsinKO cells had significantly more ER/NE-associated K48-Ub upon heat shock (Figure 4D). Thus, when a global increase in ubiquitin is provoked, TorsinKO cells sequester more K48-Ub inside blebs.

Heat shock results in the ubiquitylation and degradation of many proteins that are otherwise relatively stable [55,56]. One possibility is that blebs stabilize proteins that are targeted for degradation, as we have demonstrated for the model substrate Δ133-ORF10 [18]. While MLF2 is not directly ubiquitylated inside blebs [15], we found that MLF2-HA undergoes degradation in WT cells exposed to heat shock (Figure 4E). Note that p97 is included as a loading control in this immunoblot (Figure 4E). In contrast to WT HeLa cells under stress, MLF2-HA is stabilized in TorsinKO cells under heat shock conditions with a steady state abundance that is relatively unchanged from unstressed to stressed conditions (Figure 4E).

To determine whether K48-Ub accumulation inside blebs upon heat shock also depends on YOD1 activity, we expressed the dominant-negative YOD1-CS-FLAG construct and exposed cells to heat shock (Figure 4F,G). In untransfected heat-shocked TorsinKO cells, an intense K48-Ub signal was detected within NE foci by IF (Figure 4F, yellow arrowhead). However, upon expression of YOD1-CS-FLAG, this NE accumulation was inhibited (Figure 4F, blue arrow). We quantified this effect by determining the percent of K48-Ub foci that localize to the nuclear rim under heat shock with or without YOD1-CS-FLAG overexpression (Figure 4G). This analysis demonstrated that upon heat shock, TorsinKO cells overexpressing YOD1-CS developed significantly fewer NE-associated K48-Ub foci (Figure 4G).

As WT YOD1 overexpression also significantly reduced the amount of K48-Ub inside blebs (Figure 3A,B), we expressed WT YOD1-FLAG in WT and TorsinKO cells undergoing heat shock stress (Figure 4H). These cells were fractionated into ER/NE enrichments and analyzed by immunoblot with antibodies against K48-Ub, calnexin as a loading control, and FLAG (Figure 4H). As described above, untransfected TorsinKO cells accumulated a significant amount of ER/NE-associated K48-Ub upon heat shock (Figure 4H). However, when WT YOD1-FLAG was expressed, this heat shock-specific increase in K48-Ub was significantly reduced (Figure 4H). These results are consistent with a model wherein Torsin-deficient cells sequester ubiquitylated proteins that are—to a large extent—clients of the p97 machinery and heat labile.

## 3. Discussion

The conserved cellular phenotype observed across phylogenetically distinct Torsin ATPase loss-of-function models is NE blebs that are stalled nuclear pore complexes (Figure 1A). These blebs contain specific components including FG-nucleoporins [15,17], chaperones [18,57], MLF2 [15], and K48-Ub [14,17]. We report that cells devoid of Torsin function sequester K48-Ub protein in a p97-dependent manner (Figure 1B,C). The accumulation of ubiquitylated proteins inside blebs is only slightly decreased upon knockdown of the Ufd1/Npl4 heterodimer (Figure 2A,B), a major adaptor that recruits p97 to ubiquitylated substrates during ERAD. This suggests that the relevant p97 activity is not critically dependent on Ufd1/Npl4. Similarly, neither depletion of canonical Ub ligases implicated in ERAD (Appendix A) nor a block of de novo protein synthesis (Figure 2E) significantly affects Ub deposition in NE blebs to the extent of p97 inhibition (Figure 1B,C). Unexpectedly, depletion of the p97 adaptor UBXD1, that has to our knowledge not been tied to nuclear processes, leads to a drastic reduction in Ub accumulation in NE blebs (Figure 3A,B).

We demonstrate that globally increasing the number of ubiquitylated proteins by heat shock causes significantly more K48-Ub to be trapped inside blebs (Figure 4). This finding may argue against a specific enrichment of K48-Ub proteins and suggest a more general degradation defect exists in Torsin-deficient cells. Another interpretation consistent with this result is that blebs harbor specific heat-labile proteins that become defective and ubiquitylated under heat shock conditions. Future studies are warranted to distinguish between these two possibilities.

Our findings reported herein are consistent with a model wherein Torsin-deficient cells fail to efficiently degrade ubiquitylated proteins and instead sequester this cargo into NE herniations. We have previously demonstrated that Δ133-ORF10, a short-lived virus-derived model substrate, is rapidly degraded in WT cells but is stabilized in TorsinKO cells where it is sequestered into blebs [18]. In the present study, we generalize this finding in the physiological context of heat shock. We demonstrate that endogenous, ubiquitylated cargo provoked by heat shock is sequestered into blebs. In further support of the concept that cargo are stabilized within blebs, we find that while MLF2 is turned over in WT cells exposed to heat shock, it is strongly stabilized in TorsinKO cells under heat shock (Figure 4E). As MLF2 and Δ133-ORF10 are tightly sequestered inside blebs in TorsinKO cells, these data are consistent with the idea that proteins normally efficiently degraded in WT cells are sequestered and consequently stabilized inside blebs in Torsin-deficient cells.

Given that DYT1 dystonia is a neurological disease, a general defect in the ability to turn over potentially aberrant proteins would disproportionally affect postmitotic neurons as this cell type is unable to dilute harmful species though cell division. As this cell type is particularly vulnerable to TorsinA mutation [19,58,59], this defect may at least partially explain why TorsinA mutation exclusively causes a neurological disease. Therefore, we propose that the p97/UBXD1 axis represents a potential therapeutic target for DYT1 dystonia as its inhibition reduces K48-Ub accumulation inside blebs.

As Torsin ATPases localize within the ER/NE system [3,60,61], the protein quality control defect arising upon Torsin loss of function may be related to ERAD. While the involvement of Torsin activity in ERAD has been investigated [14,62], it has not been found to play a critical role in processing many canonical ERAD model substrates. Furthermore, Torsin depletion is not consistently associated with causing general ER stress [14,17,63]. As we report herein, inhibiting translation (Figure 2E), depleting the Ufd1/Npl4 heterodimer (Figure 2A,B) or canonical Ub ligases required for ERAD (Appendix A) fail to significantly reduce the amount of K48-Ub sequestered into blebs. Thus, if Torsin loss of function results in a PQC defect related to ERAD, it would likely affect clients of a distinct pathway that are retrotranslocated and ubiquitylated by a noncanonical mechanism. It is furthermore unlikely that these clients would be newly synthesized proteins given that CHX treatment has little impact on the K48-Ub cargo inside blebs (Figure 2E).

Our data suggest that the p97 interaction partners UBXD1 and YOD1 function in the PQC pathway that becomes dysregulated upon Torsin deficiency. While YOD1 is best characterized as participating in ERAD-related processes [50,64], it has also been reported to facilitate autophagy of damaged lysosomes [47]. Notably, this latter function is in conjunction with UBXD1 and the p97 adaptor PLAA [47]. While we did not find that knockdown of PLAA effected K48-Ub levels inside blebs (data not shown), UBXD1 has been reported to function in a wide range of processes including mitophagy [48] and trafficking [46]. Thus, it will be interesting for future endeavors to test whether UBXD1 and YOD1 contribute to the same p97-dependent process in Torsin-deficient cells or if these two adaptors function with p97 in distinct, dysregulated processes.

The small-molecule p97 inhibitor CB-5083 depletes the K48-Ub cargo inside blebs within four hours of treatment (Figure 1B). This relatively short treatment time reveals a previously unappreciated feature of blebs: K48-Ub cargo must undergo some dynamic exchange with the nucleoplasm. This dynamic feature of blebs has not been detected by previous workflows monitoring steady-state levels of K48-Ub sequestration [14,15,17,58]. It will be important to determine the fate of the K48-Ub protein once they are released from blebs under p97 depletion conditions. For example, this cargo could be degraded once inside the nucleoplasm or it could persist within the cell and pose an even greater proteotoxic threat, depending on the resident times in NE blebs versus the nucleoplasm. Efforts to uncover how p97 inhibition affects blebs may also reveal whether proteasome flux is impaired in models of DYT1 dystonia and the identity of K48-Ub conjugates. Answering these questions will not only advance therapeutic options but further our understanding of Torsin ATPases.

## 4. Materials and Methods

### 4.1. Antibodies

The following antibodies were used in this study (WB, Western blot (Bio-Rad [Hercules, CA, USA]). IF, immunofluorescence): K48 linkage-specific polyubiquitin (WB, 1:4000. IF, 1:500. MilliporeSigma [Burlington, MA, USA], Apu2), HA-peptide (WB, 1:2000. IF, 1:500. Roche, 3F10), p97 (WB, 1:7000. Abcam [Cambridge, England], ab109240), α-tubulin (WB, 1:5000. MilliporeSigma [Burlington, MA, USA], T5168), α-GAPDH (WB, 1:10,000. Proteintech [Rosemont, IL, USA], 60004-1-Ig), α-Ufd1 (WB, 1:1000. Cell Signaling [Danvers, MA, USA], 13789), α-Npl4 (WB, 1:1000. Cell Signaling, 13489), α-UBXD1 (WB, 1:500. Bethyl Laboratories [Montgomery, TX, USA], A302-931A), α-hnRNPA1 (WB, 1:2000. Abcam [Cambridge, England], ab5832), α-emerin (WB, 1:4000. Cell Signaling [Danvers, MA, USA], 30853S), α-calnexin (WB, 1:2000. Abcam [Cambridge, England], 75801), FLAG peptide (WB, 1:4000. IF, 1:500. MilliporeSigma [Burlington, MA, USA], F3165), rabbit IgG HRP conjugate (WB, 1:10,000. SouthernBiotech [Birmingham, AL, USA], 4030-05), mouse IgG HRP conjugate (WB, 1:20,000. SouthernBiotech [Birmingham, AL, USA], 1030-05), rabbit and mouse IgG Alexa488 conjugates (IF, 1:700. Invitrogen [Waltham, MA, USA], A11008 and A28175), and rabbit and mouse IgG Alexa568 (IF, 1:700. Invitrogen [Waltham, MA, USA], A-11011 and A-11004).

### 4.2. Cell Culture and Cell Lines

Torsin-deficient [17] and WT HeLa cells were maintained in Dulbecco’s Modified Eagle’s Medium (DMEM) supplemented with 10% fetal bovine serum (Thermo Fisher Scientific [Waltham, MA, USA]) and 100 units mL^−1^ of penicillin-streptomycin (Thermo Fischer Scientific [Waltham, MA, USA]). Cells were verified to be free of mycoplasma contamination through the absence of extranuclear Hoechst 33342 (Life Technologies [Carlsbad, CA, USA]) staining. Heat shock was achieved by incubating cells for 16 h at 42 °C.

### 4.3. Small-Molecule Treatment, Plasmids, and Transient Transfections

Inhibition of p97 was achieved using the small-molecule CB-5083 [24,25] (Apexbio Technology [Houston, TX, USA], 1542705-92-9). The compound was dissolved in DMSO to a stock concentration of 10 mM. The stock solution was diluted 1:2000 in DMEM completed as described above for a final concentration of 5 µM. Cells were exposed to media containing CB-5083 for four hours prior to harvesting for downstream applications.

The cDNA sequence encoding MLF2-HA was cloned into the pcDNA3.1+ vector as previously described [15] using standard PCR. WT YOD1-FLAG, YOD1-CS-FLAG, and p97-QQ constructs in pcDNA3.1+ were cloned as previously described [50]. HA-tagged and untagged LBR-1600* were cloned as previously described [28,29]. The cDNA encoding UBXD1 was amplified from HeLa cell cDNA and cloned into pcDNA3.1+ using NheI and XbaI with an N-terminal FLAG tag. pENTR-AIF (AIFM1) was a gift from Huda Zoghbi (Addgene plasmid #16182; http://n2t.net/addgene:16182, accessed on 20 January 2019; RRID: Addgene_16182) and was subcloned into pcDNA3.1+ with a C-terminal FLAG tag.

Transient plasmid transfections were performed using Lipfectamine 2000 (Invitrogen [Waltham, MA, USA]) according to the manufacturer’s instructions. Constructs were allowed to express for 24 h prior to analyses.

### 4.4. siRNAs, Transient RNAi Knockdowns, and qPCR Validation

The following siRNAs [47] targeting UBXD1 were ordered from MilliporeSigma [Burlington, MA, USA]. Forward oligo 1: 5′-CCAGGUGAGAAAGGAACUU[dT][dT]-3′. Forward oligo 2: 5′-UCAGAUACCACGUUGGUCCC[dT][dT]-3′. Nontargeting and siRNAs targeting Ufd1, Npl4, p47, Hrd1, MARCH6, and gp78 were purchased from Horizon Discovery [Waterbeach, England] as SMARTpools.

RNAi knockdowns were performed with Lipfectamine RNAiMAX (Invitrogen [Waltham, MA, USA]) according to the manufacturer’s instructions and allowed to knock targets down for 48 h before analyses.

Knockdown efficiency was validated by quantitative PCR (qPCR) using iQ SYBR Green (Bio-Rad [Hercules, CA, USA]) mix with a CFX Real-Time PCR 639 Detection System (Bio-Rad [Hercules, CA, USA]). Each knockdown was evaluated using the ΔΔCt method using the internal control transcript RPL32. Primer sequences used for qPCR were as follows: UBXD1 (FWD: TGGAGAGGCACAAGGAACAGC, REV: CCCGCTTGATCTCCTCTGCT), Ufd1 [65] (FWD: GAGGGAAGATAATTATGCCAC, REV: CTTCCAAGAGTAAGTTCTGC), Npl4 (FWD: GCTTGGCCACCTATTTGTCTCAGAA, REV: CATTGGTGACCAGGAACAGCAAGA), Hrd1 [66] (FWD: GCGAGACATGATGGCATCTG, AACCCCTGGGACAACAAGG, REV: GCGAGACATGATGGCATCTG), MARCH6 (FWD: AGCATGCTCGAAATAACAACGCT, REV: GGCGGTAAGGCTGAAAGCCA), gp78 (FWD: CGTGTGCCACTGGACCTCAG, REV: CACCAGCATGCGCTGTCTCT), p47 (FWD: AGTACCAGCTCTCCAGCCCAA, REV: CGCCGTCTGCAAGCCGAA). All primers were synthesized by Integrated DNA Technologies [Coralville, IA, USA].

### 4.5. Immunofluorescence and Confocal Microscopy

HeLa cells were grown on coverslips prior to processing for IF. Cells were fixed onto coverslips in 4% paraformaldehyde (Thermo Fisher Scientific [Waltham, MA, USA]) for 20 min at room temperature, then permeabilized in 0.1% Triton X-100 (MilliporeSigma [Burlington, MA, USA]) for 10 min. After blocking in 4% bovine serum albumin for 15 min, cells were incubated for 45 min with primary antibodies diluted 1:500 in blocking solution. After exhaustive washing in phosphate buffered saline (PBS), cells were incubated in fluorescent secondary antibodies diluted 1:700 in blocking solution. Prior to mounting onto slides, the cells were stained with Hoechst 33342 (Life Technologies [Carlsbad, CA, USA]) and adhered to slides with Fluoromount-G (Southern Biotech [Birmingham, AL, USA]).

Confocal images were collected on an LSM 880 laser scanning microscope (Zeiss [Jena, Germany]) with a C Plan-Apochromat 63×/1.40 oil DIC M27 objective using ZEN 2.1 software (Zeiss [Jena, Germany]). Image quantification and processing was performed in Fiji [67] or CellProfiler [68] as described below.

### 4.6. Immunoprecipitation and Immunoblot Analysis

All immunoprecipitation (IP) experiments were conducted under native conditions. Cells were transfected or exposed to relevant conditions 16–24 h prior to harvesting for IP, then lysed in NET buffer (150 mM NaCl, 50 mM Tris pH 7.4, 0.5% NP-40) supplemented with EDTA-free protease inhibitor cocktail (Roche [Basel, Switzerland]) and 5 mM NEM (MilliporeSigma [Burlington, MA, USA]). Equal amounts of protein were loaded onto protein A beads conjugated to 1 µg anti-p97 (Abcam [Cambridge, England], ab109240) or Anti-FLAG M2 Affinity Gel (MilliporeSigma, [Burlington, MA, USA], A2220). After incubating for three hours, the resin was washed extensively in NET buffer and protein was eluted in SDS reducing buffer for downstream immunoblot analyses.

Cell lysates were prepared for immunoblot in NET buffer plus protease inhibitors and 5 mM NEM as described above. Immunoblotting with IP eluates or cell lysates was performed with SDS-PAGE gels (Bio-Rad [Hercules, CA, USA]) and transferred onto PVDF membranes (Bio-Rad [Hercules, CA, USA]). Membranes were blocked in 5% *w*/*v* milk in PBS + 0.1% Tween-20 (MilliporeSigma [Burlington, MA, USA]). Primary and HRP-conjugated secondary antibodies were diluted in blocking buffer. Blots were visualized by chemiluminescence on a ChemiDoc Gel Imaging System (Bio-Rad [Hercules, CA, USA]).

### 4.7. Cycloheximide Chase

TorsinKO HeLa cells were plated in a 10 cm dish and transfected with HA-LBR 1600* [28], which was allowed to express for 24 h prior to harvesting cells. After 24 h, the cells were either trypsinized and split into two tubes for immunoblot or kept within the culture dish containing coverslips for IF. Tubes for immunoblot were incubated at 37 °C with gentle shaking and treated with either DMSO or 100 µg/mL CHX diluted in completed DMEM described above. Culture dishes for IF were treated with the same conditions and remained within the culturing incubator. Aliquots for immunoblot or coverslips were taken at zero, one, two, five, or seven hours post treatment. Cells were collected via centrifugation and subjected to immunoblot or fixed with 4% paraformaldehyde and processed for IF as described above.

### 4.8. NE Enrichment

NE fractions were enriched from whole-cell lysates as previously described [29]. Cells were collected by centrifugation and resuspended in buffer A (10 mM HEPES, pH 7.4, 250 mM sucrose, 2 mM MgCl2) supplemented with EDTA-free protease inhibitor cocktail (Roche [Basel, Switzerland]). Cell pellets were homogenized in 100 µL of buffer A by passing through a 25-gague needle and layered on top of 10 mL STM 0.9 buffer (50 mM Tris, pH 7.4, 0.9 M sucrose, 5 mM MgCl_2_). The homogenates were centrifuged through the STM 0.9 layer at 1000× *g* for 10 min twice and a white pellet was observed at the bottom. This pellet, composed of the nuclei/ER and cell debris, was resuspended in TP buffer (10 mM Tris, pH 8.0, 10 mM Na_2_HPO_4_, 2 mM MgCl_2_) supplemented with heparin (2 mg/mL), benzonase nuclease (2 μL/mL), and protease inhibitor cocktail (Roche [Basel, Switzerland]) and rotated overnight at 4 °C. In the morning, the nuclei were spun at 15,000× *g* for 45 min to separate the ER/NE (pellet) from the nucleoplasm (supernatant). ER/NE enrichments were resuspended in 1% SDS and prepared for immunoblot analysis.

### 4.9. Image Processing and Statical Analysis

All image quantification was performed using Fiji [67] or CellProfiler [68] software. The number of NE-associated K48-Ub foci was determined with Fiji software by defining a region of interest (ROI) around the nucleus of individual cells and quantifying the number of K48-Ub foci using the “Find Maxima” function. In line with our previous publications [15,17,18], the prominence or noise tolerance was set to 10. All statistical analyses for quantifying NE-associated K48-Ub foci utilized the Mann–Whitney U test, which does not assume the distributions are normal.

The intensity of K48-Ub in TorsinKO cells at 37 and 42 °C was assessed using CellProfiler [68] software. Cells were processed for IF as described above and imaged for K48-Ub using identical exposure times for each condition. For comparative analysis of K48-Ub intensity at 37 and 42 °C, the integrated intensity units of K48-Ub foci at the nuclear rim of each cell were quantified in CellProfiler. For determination of the distribution of K48-Ub at 42 °C in the absence and presence of YOD1-CS-FLAG, CellProfiler was used to determine the fraction of total K48-Ub in the outer nuclear region for each cell. The populations were statistically analyzed using the Mann–Whitney U test, which does not require normal distributions.

## Figures and Tables

**Figure 1 ijms-23-04627-f001:**
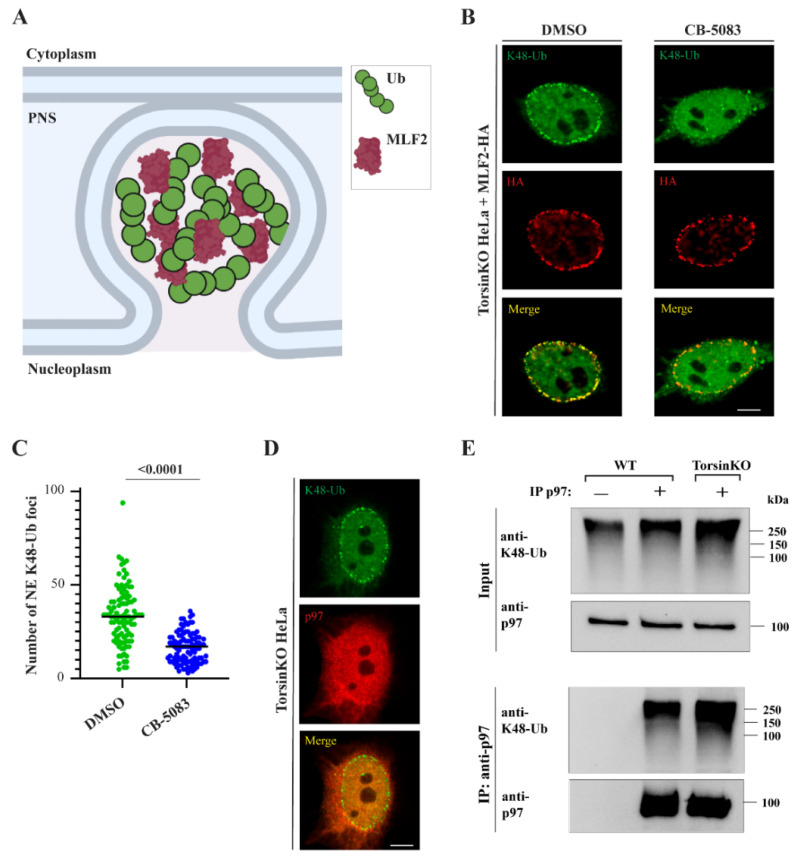
K48-Ub accumulation inside NE blebs of Torsin-deficient cells requires p97 activity. (**A**) Schematic diagram of a NE bleb. PNS, perinuclear space. Ub, ubiquitin. (**B**) Representative immunofluorescence (IF) images of TorsinKO HeLa cells overexpressing MLF2-HA under vehicle (DMSO) or p97 inhibition (CB-5083) conditions. Cells were treated with 5 µM CB-5083 for four hours before processing for IF, stained with K48-Ub and HA antibodies, and imaged by confocal microscopy. Scale bar, 5 µm. (**C**) The number of NE-associated K48-Ub foci in TorsinKO cells upon DMSO or CB-5083 treatment was quantified for 100 cells/condition. Statistical analysis was performed using a Mann–Whitney U test. (**D**) Representative IF image of p97 localization in TorsinKO cells. Cells were processed for IF and stained with p97 and K48-Ub antibodies. Scale bar, 5 µm. (**E**) An immunoprecipitation (IP) of endogenous p97 under native conditions from WT and TorsinKO cells. The IP was analyzed by immunoblot with antibodies against p97 and K48-Ub.

**Figure 2 ijms-23-04627-f002:**
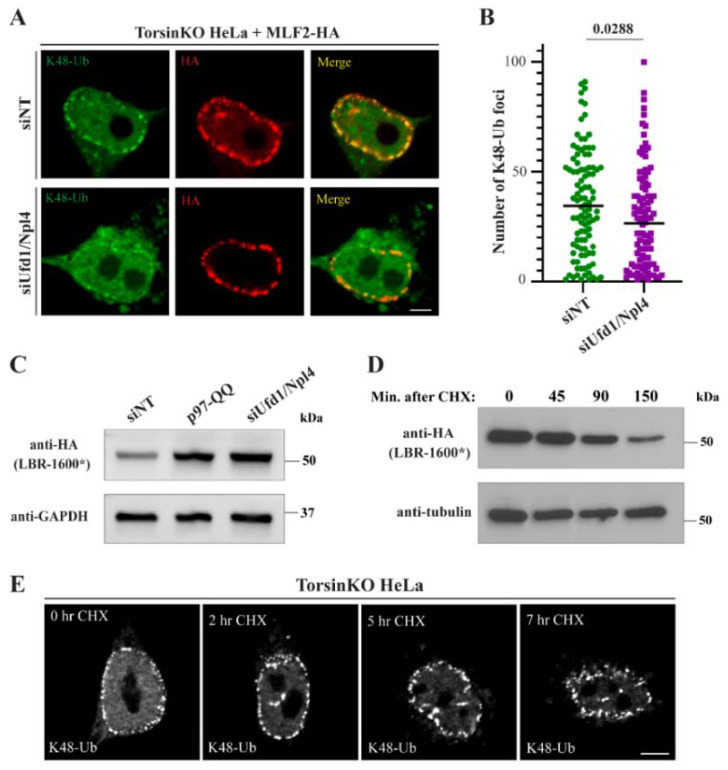
The K48-Ub proteins sequestered inside blebs do not result from canonical ERAD. (**A**) Representative IF images of TorsinKO cells overexpressing MLF2-HA under nontargeting (siNT) or Ufd1/Npl4 knockdown conditions. Cells were treated with 25 nM RNAi against both Ufd1 and Npl4 for 48 h before processing for IF. Cells were stained with antibodies against K48-Ub and HA. Scale bar, 5 µm. (**B**) The number of NE-associated K48-Ub foci in TorsinKO cells under nontargeting or Ufd1/Npl4 knockdown conditions was quantified for 100 TorsinKO cells/condition. Statistical analysis was performed using a Mann–Whitney U test. (**C**) Representative immunoblot of the ERAD substrate LBR-1600* in TorsinKO cells under siNT, siUfd1/Npl4, or p97-QQ overexpression conditions. p97-QQ is a dominant-negative mutation that inhibits endogenous p97 activity. The Ufd1/Npl4 knockdown was conducted for 48 h while LBR-1600* and p97-QQ were expressed for 24 h prior to immunoblotting. (**D**) TorsinKO cells expressing the short-lived HA-LBR-1600* were treated with CHX for the indicated timepoints and processed for immunoblot. Cells were allowed to express HA-LBR-1600* for 24 h before treatment with 100 µg/mL CHX. (**E**) Representative IF images of TorsinKO cells treated with CHX for the indicated timepoints. Cells were treated with 100 µg/mL of CHX prior to processing for IF and stained with an K48-Ub antibody. Scale bar, 5 µm.

**Figure 3 ijms-23-04627-f003:**
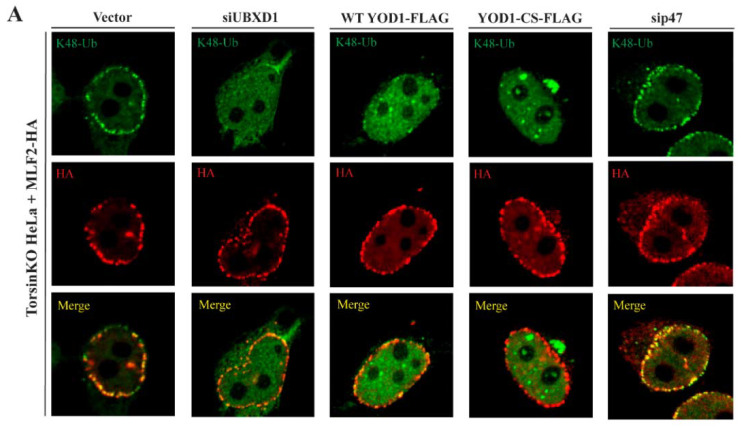
p97 depends on the cofactors YOD1 and UBXD1 to generate the K48-ubiquinated proteins sequestered inside blebs. (**A**) Representative IF images of TorsinKO cells overexpressing MLF2-HA under conditions of normal or disrupted p97 activity. p97 activity was disrupted by knocking down UBXD1 or p47 or overexpressing the dominant-negative YOD1-CS-FLAG. UBXD1 and p47 were knocked down for 48 h and overexpressed constructs were overexpressed for 24 h prior to processing for IF. Cells were stained with antibodies against K48-Ub and HA. Note that WT YOD1-FLAG is an active deubiquitinase (DUB) that cleaves the K48-Ub chains off clients that are normally ubiquitylated and sequestered into blebs. Scale bar, 5 µm. (**B**) The number of K48-Ub foci was quantified for 100 TorsinKO cells/condition. Statistical analysis was performed using a Mann–Whitney U test. N.s., not significant. (**C**) An immunoblot confirming the effects of siRNA targeting Ufd1, Npl4, or UBXD1 in TorsinKO cells after 48 h of knockdown. (**D**) An IP of FLAG-tagged WT YOD1, YOD1-CS, UBXD1, or AIMF1 from TorsinKO cells expressing MLF2-HA. Cells were allowed to express the constructs for 24 h prior to harvesting. An IP of the FLAG tag for all samples was analyzed by immunoblot against the FLAG and HA tags. MLF2-HA specifically co-immunoprecipitates with WT YOD1, UBXD1, and AIFM1.

**Figure 4 ijms-23-04627-f004:**
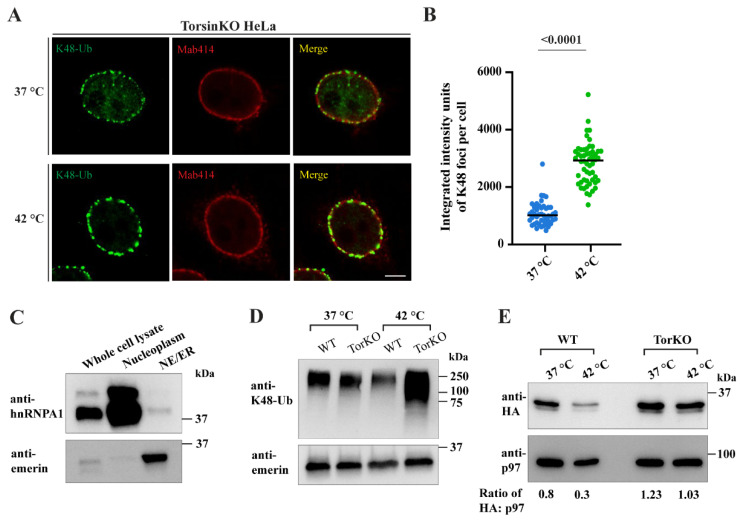
NE blebs in Torsin-deficient cells sequester ubiquitylated proteins that are produced by heat shock stress. (**A**) Representative IF images of NE-associated K48-Ub foci in TorsinKO cells upon heat shock. Cells were cultured at 37 °C or 42 °C for 16 h before processing for IF. Cells were stained with an antibody against K48-Ub and Mab414, which recognizes a subset of FG-nucleoporins. Scale bar, 5 µm. (**B**) Quantification of the average intensity of K48-Ub signal inside blebs from TorsinKO cells exposed to 37 °C or 42 °C for 16 h. This value was quantified for 50 TorsinKO cells/condition. Statistical analysis was performed using a Mann–Whitney U test. (**C**) An immunoblot demonstrating the subcellular fractionation of TorsinKO cells. Cells were enriched for NE/ER fractions (indicated by the emerin antibody) that were free from significant nucleoplasmic contamination (indicated by the anti-hnRNPA1 antibody). (**D**) NE/ER fractions from WT or TorsinKO HeLa cells under unstressed or heat shock stress conditions were analyzed by immunoblot with antibodies against K48-Ub and emerin. Cells were exposed to a 42 °C heat shock for 16 h before harvesting. (**E**) Unstressed or heat-shocked WT and TorsinKO cells overexpressing MLF2-HA for 24 h were fractionated into ER/NE enrichments. The fractions were analyzed by immunoblot using. Note that p97 controls for protein loading in this immunoblot. The ratio of HA to p97 density was calculated for each lane. (**F**) Representative IF images of TorsinKO cells expressing YOD1-CS-FLAG under heat shock stress as described in panel (**A**). Cells were transfected with YOD1-CS-FLAG eight hours prior to the beginning of heat shock. Blue arrow, transfected cell. Yellow arrowhead, untransfected cell. Scale bar, 5 µm. (**G**) Quantification of the percent of total K48-Ub signal that localize to the nuclear rim in TorsinKO cells upon 42 °C heat shock with or without YOD1-CS-FLAG overexpression. This value was obtained for 30 cells/condition. Statistical analysis was performed using a Mann–Whitney U test. (**H**) WT and TorsinKO cells expressing WT YOD1-FLAG for 24 h were subjected to heat shock stress and enriched for ER/NE fractions. These fractions were compared by immunoblot to unstressed or heat-shocked cells in the absence of overexpressed YOD1-FLAG.

## Data Availability

All raw data used to generate plots and original scans of immunoblots are available upon request.

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
