# Peer review of "p97/UBXD1 Generate Ubiquitylated Proteins That Are Sequestered into Nuclear Envelope Herniations in Torsin-Deficient Cells"

_ijms, 2022, doi:10.3390/ijms23094627_

Round 1
Reviewer 1 Report
The manuscript entitled "p97 / ULXD1 generates ubiquitinated proteins that are sequestered in the nuclear envelope ..." by Prophet S.M. et al, provides new data regarding the mechanism of development of DYT1 dystonia. This work clearly and convincingly shows that the p97 / UBXD1 axis is an important regulatory factor for the development of this disease caused by Torsin ATPase disfunction. The authors also postulate that it is not excluded that p97 could be a molecular target for targeted therapy, which would be important to improve the lives of patients.
The manuscript is solidly written. The presented data clearly proves that the thesis and conclusions contained in this paper are true and relevant.
In my opinion it should be accepted for publication in IJMS as it is now.
Author Response
We thank the reviewer for a timely and supportive review.
Reviewer 2 Report
Prophet and co-workers describe in their manuscript entitled “p97/UBXD1 generate ubiquitinated proteins that are sequestered into nuclear envelope herniations in Torsin-deficient cells” a series of studies performed with Torsin-deficient HeLa cells that the group reported in a 2016 MBoC paper. These cells are characterised by the presence of numerous nuclear envelope blebs that accumulate Lys48-linked polyubiquitin (K48-Ub) chains. By using a specific inhibitor of the p97 ATPase, the authors found that K48-Ub levels were significantly reduced in the blebs. Moreover, they demonstrated an interaction between p97 and K48-Ub by immunoprecipitation with anti-p97 antibodies. Next, the authors tested the requirement for different p97 co-factors and concluded that YOD1 and UBXD1 but not Ufd1/Npl4 are needed. Finally, the authors performed heat stress experiments to demonstrate an increase in K48-Ub levels in the nuclear envelope/ER fraction specifically in Torsin-deficient cells.
The study is interesting, convincing in most aspects and well-written. However, the following important points should be corrected or clarified prior to publications.
The Laudermilch et al, 2016 paper states that the human genome encodes four Torsin ATPases and describes the generation of a quadruple KO. Is this the cell line that was used in this study? The current study states that the human genome encodes five Torsin ATPases (line 32). What is the correct number? Is the cell line used in this study depleted for all Torsin genes?
Information on number of replicas is missing. This is particularly relevant for results without quantifications such as those represented in Figure 4.
The authors conclude that Udf1/Npl4 is not important for the p97-mediated increase in K48-Ub because the decrease in Figure 2A-B is only modest. However, the effect by Udf1/Npl4 depletion is equally modest in the control experiment in Figure 2C-D, which casts doubt on the authors’ conclusion.
Rather than representing siRNA efficiencies by qPCR, it would be relevant to show protein levels by Western blotting.
Based on IF data in Figure 1D, the authors write that “Thus, p97 activity at locations distinct from the bleb appears to be required for the K48-Ub accumulation in Torsin-deficient cells”. However, p97 seems to be present everywhere, including in blebs. Hence, its activity might take place inside the blebs. Please show high-mag images of the blebs.
The correlation between images in Figure 3A and quantifications in Figure 3B is incomplete (no images of sip47 in 3A; no quantification of p97-QQ in 3B).
The authors write that “In further support of this concept, we find that while MLF2 is turned over in WT cells exposed to heat shock, it is strongly stabilized in Torsin KO cells under heat shock (Figure 4D).” However, controls for equal loading are missing and the effects should be quantified.
Minor details:
In line 199, “Figure 2C” should be “Figure 2E”.
In line 201, “Figure 2D” should be “Figure 2F”.
Lines 296-297: Maybe data was normalised to RPL32 but they are represented as relative to the siNT control.
In the experiments shown in Figure 3, why was p97 inhibited with a dominant-negative p97-QQ rather than the drug that was used successfully in Figure 1?
Author Response
We appreciate the positive and timely evaluation of our manuscript by both reviewers and the constructive criticism that has led to a considerably improved manuscript. As detailed in our point-by-point response below, we have constructively addressed all points of concern in this revised version. The results are not only in excellent agreement with our original conclusions, but also allow us to make a significantly more convincing case for a specific role of the p97/UFD1 axis in the context of Ubiquitin accumulation. We hope that the reviewers and editor agree that the revised manuscript is now acceptable for publication. Please note our responses to each point raised by reviewer 2 (in bold) are italicized.
1) The Laudermilch et al, 2016 paper states that the human genome encodes four Torsin ATPases and describes the generation of a quadruple KO. Is this the cell line that was used in this study? The current study states that the human genome encodes five Torsin ATPases (line 32). What is the correct number? Is the cell line used in this study depleted for all Torsin genes?
We thank the reviewer for pointing this out. We employ the quadruple KO cells deficient of TorsinA, B, 2A, and 3A as described in Laudermilch et al. 2016 in this manuscript. The text has been updated to clarify this point. Note that the “fifth” Torsin4A is a poorly conserved outlier that has not been characterized in the published literature (to our knowledge).
- Information on number of replicas is missing. This is particularly relevant for results without quantifications such as those represented in Figure 4.
In the revised version, we have ensured that all figure legends describe the sample size quantified in the corresponding panels. Specifically, we agree that Figure 4 lacks sufficient quantification. To address this, we have quantified the average K48-Ub intensity within NE blebs for n = 50 cells at either 37ºC or 42ºC. These new data accompany representative IF and immunoblot data (Figure 4B).
In addition to quantifying K48-Ub intensity, we have robustly quantified the number of NE-associated K48-Ub foci that accumulate at 42ºC upon overexpression of YOD1-CS-FLAG for 30 cells (Figure 4G).
- The authors conclude that Udf1/Npl4 is not important for the p97-mediated increase in K48-Ub because the decrease in Figure 2A-B is only modest. However, the effect by Udf1/Npl4 depletion is equally modest in the control experiment in Figure 2C-D, which casts doubt on the authors’ conclusion.
We thank the reviewer for making this important point. We agree that the effect of siUfd1/Npl4 on TCR-α stabilization was modest. Upon further investigation, we found that TCR-α was not an ideal control substrate for our system as it was only poorly expressed. We have therefore utilized LBR-1600*, a short-lived mutant of the inner nuclear membrane protein LaminB receptor, as this is the more suitable control due to its localization to the nuclear envelope. Note that our laboratory has shown that LBR-1600* is rapidly degraded in a proteasome- and p97-dependant manner (10.7554/eLife.16011, 10.1016/bs.mie.2018.12.033).
We now report that knocking down Ufd1/Npl4 stabilizes LBR-1600*to the same extent (!) as p97 inhibition by overexpression of the dominant-negative p97-QQ (Figure 2C). Thus, we can now robustly conclude that while knocking down Ufd1/Npl4 significantly stabilizes substrates destined for degradation, it does not strongly affect K48-Ub accumulation within blebs. Therefore, the K48-Ub clients inside blebs do not appear clients of the canonical ERAD machinery.
- Rather than representing siRNA efficiencies by qPCR, it would be relevant to show protein levels by Western blotting.
We agree with the reviewer and have replaced qPCR data with immunoblots for the relevant knockdowns that are germane for the main conclusions of our study in the main figure (Figure 3D). The original qPCR data has been moved to supplemental figure 2B.
- Based on IF data in Figure 1D, the authors write that “Thus, p97 activity at locations distinct from the bleb appears to be required for the K48-Ub accumulation in Torsin-deficient cells”. However, p97 seems to be present everywhere, including in blebs. Hence, its activity might take place inside the blebs. Please show high-mag images of the blebs.
We have performed a number of studies (10.1091/mbc.E16-07-0511, 10.1083/JCB.201910185) wherein the presence of p97 inside blebs was addressed. In none of these published studies, nor in the present one, do we find p97 to be enriched inside blebs. (Note that 10.1083/JCB.201910185 includes zoomed-in IF images of the nuclear rim with anti-p97 staining). High abundance proteins that are enriched in blebs, for example HSP70s, re-localize from diffusely throughout the cell to a nuclear rim punctate distribution (10.1101/2021.10.26.465916). Thus, if p97 was stably enriched within the bleb, this would almost certainly be observed by our IF studies. Please note also that we have never identified this protein as enriched in blebs by mass spectrometry.
We agree with the reviewer, however, that it is formally possible p97 is at least transiently associated with the bleb lumen, or might exert an effect while being present below the detection limit of our readouts. We have therefore modified the sentence regarding p97 acting at locations distinct from blebs to read “Consistent with our previous observations {Laudermilch, 2016 #2;Rampello, 2020 #2535}, we found it was not stably enriched within blebs (Figure 1D) although we cannot exclude the formal possibility that p97 is present at levels below the detection limit of immunofluorescence (IF). While it is possible that p97 acts directly within the bleb lumen to generate the K48-Ub proteins, p97 activity at locations distinct from the bleb may also be required for the K48-Ub accumulation in Torsin-deficient cells.”
- The correlation between images in Figure 3A and quantifications in Figure 3B is incomplete (no images of sip47 in 3A; no quantification of p97-QQ in 3B).
We thank the reviewer for pointing out this discrepancy. We have elected to remove the image of p97-QQ as it is redundant with the data from Figure 1. In its place, we have included images of the sip47 condition. The quantifications in panel 3B are now consistent with panel 3A.
- The authors write that “In further support of this concept, we find that while MLF2 is turned over in WT cells exposed to heat shock, it is strongly stabilized in Torsin KO cells under heat shock (Figure 4D).” However, controls for equal loading are missing and the effects should be quantified.
While we intended p97 to act as a loading control for the immunoblot shown in Figure 4D (now panel 4E), we agree this was not clearly described. We have modified the text to include clarifying statements to the results and figure legend.
In addition, we have used densitometry to quantify the stabilization effect and include the values in the new panel 4E.
Minor details:
In line 199, “Figure 2C” should be “Figure 2E”.
We thank the reviewer for catching this error. It has been corrected.
In line 201, “Figure 2D” should be “Figure 2F”.
We thank the reviewer for catching this error. It has been corrected.
Lines 296-297: Maybe data was normalised to RPL32 but they are represented as relative to the siNT control.
To clarify our rationale for standardization: the ΔΔCt method involves normalizing the raw Ct values for each transcript to RPL32 as an internal “loading” control for the amount of cDNA in each reaction. Without this correction it would not be possible to compare Ct values across different samples. Thus, this normalization is taken into account prior to normalizing each value to the siNT control (hence the detla-delta Ct method as opposed to simply the delta Ct method). As there is no relevant comparison between the indicated transcripts and RPL32 values, nor is there one without this normalization step, we represent the qPCR data as relative to siNT.
In the experiments shown in Figure 3, why was p97 inhibited with a dominant-negative p97-QQ rather than the drug that was used successfully in Figure 1?
We agree the addition of p97-QQ in Figure 3 was redundant and muddling. We have therefore omitted this panel in order to make Figure 3 more internally consistent. This change is also in agreement with the reviewer’s previous comment regarding consistency between quantifications and IF images.

Round 2
Reviewer 2 Report
I wish to congratulate the authors on their revised manuscript, which I find very suitable for publication.